# Impact of high platelet turnover on the platelet transcriptome: Results from platelet RNA-sequencing in patients with sepsis

Thomas G. Nührenberg[1,2]*, Jasmin Stöckle[1], Federico Marini[3], Mark Zurek[1], Björn A. Grüning[4], Vladimir Benes[5], Lutz Hein[2,6], Franz-Josef Neumann[1], Christian Stratz[1¤a], Marco Cederqvist[1©], Willibald Hochholzer[1©¤b]

1 Klinik für Kardiologie und Angiologie II, Universitätsklinik Freiburg, Universitäts-Herzzentrum Campus Bad Krozingen, Bad Krozingen, Germany, 2 Institut für experimentelle und klinische Pharmakologie und Toxikologie, Medizinische Fakultät, Albert-Ludwigs-Universität Freiburg, Freiburg, Germany, 3 Institut für Medizinische Biometrie, Epidemiologie und Informatik, Universitätsmedizin der Johannes-Gutenberg-Universität Mainz, Mainz, Germany, 4 Institut für Informatik, Albert-Ludwigs-Universität Freiburg, Freiburg, Germany, 5 Genomics Core Facility, European Molecular Biology Laboratory, Heidelberg, Germany, 6 BIOSS Centre for Biological Signalling Studies, University of Freiburg, Freiburg, Germany

© These authors contributed equally to this work.
¤a Current address: Novartis Pharma AG, Basel, Switzerland
¤b Current address: Krankenhaus Juliusspital, Klinikum Würzburg Mitte, Würzburg, Germany
* thomas.nuehrenberg@uniklinik-freiburg.de

**Data Availability Statement:** The RNA-seq data presented in this paper have been deposited in the European Nucleotide Archive (ENA, https://www.

## Abstract

### Background

Sepsis is associated with high platelet turnover and elevated levels of immature platelets. Changes in the platelet transcriptome and the specific impact of immature platelets on the platelet transcriptome remain unclear. Thus, this study sought to address whether and how elevated levels of immature platelets affect the platelet transcriptome in patients with sepsis.

### Methods

Blood samples were obtained from patients with sepsis requiring vasopressor therapy (n = 8) and from a control group of patients with stable coronary artery disease and otherwise similar demographic characteristics (n = 8). Immature platelet fraction (IPF) was determined on a Sysmex XE 2100 analyser and platelet function was tested by impedance aggregometry. RNA from leukocyte-depleted platelets was used for transcriptome analysis by Next Generation Sequencing integrating the use of unique molecular identifiers.

### Results

IPF (median [interquartile range]) was significantly elevated in sepsis patients (6.4 [5.3–8.7] % vs. 3.6 [2.6–4.6] %, p = 0.005). Platelet function testing revealed no differences in adenosine diphosphate- or thrombin receptor activating peptide-induced platelet aggregation between control and sepsis patients. Putative circular RNA transcripts were decreased in platelets from septic patients. Leukocyte contamination defined by CD45 abundance levels in RNA-sequencing was absent in both groups. Principal component analysis of transcripts

ebi.ac.uk/ena/browser/home) under accession number PRJEB45262.

**Funding:** TGN and LH are members of SFB1425, funded by the Deutsche Forschungsgemeinschaft (DFG, German Research Foundation #422681845). In addition, this work was funded by internal grants of Universitäts-Herzzentrum Freiburg • Bad Krozingen and the Institute of Experimental and Clinical Pharmacology and Toxicology, Faculty of Medicine, University of Freiburg. The funders had no role in study design, data collection and analysis, decision to publish, or preparation of the manuscript.

**Competing interests:** The authors have declared that no competing interests exist.

**Abbreviations:** miRNA, microRNA; PRP, platelet-rich plasma; RNA-Seq, RNA-Sequencing; ASA, acetylsalicylic acid.

showed only partial overlap of clustering with IPF levels. RNA sequencing showed up-regulation of 524 and down-regulation of 118 genes in platelets from sepsis patients compared to controls. Upregulated genes were mostly related to catabolic processes and protein translation. Comparison to published platelet transcriptomes showed a large overlap of changes observed in sepsis and COVID-19 but not with reticulated platelets from healthy donors.

## Conclusions

Patients with sepsis appear to have a less degraded platelet transcriptome as indicated by increased levels of immature platelets and decreased levels of putative circular RNA transcripts. The present data suggests that increased protein translation is a characteristic mechanism of systemic inflammation.

## Introduction

Sepsis is the most common cause of death in hospitalised patients [1]. Large amounts of circulating pathogens trigger a massive immune response and an activation of the coagulation system. If uncontrolled, progressive dysfunction of the immune and coagulation system predisposes for multi-organ failure and death. Due to their homeostatic and immunomodulatory function, platelets play a key role in sepsis [2]. Platelet activation in sepsis has been extensively studied and has been shown to be prognostic for survival [3]. Yet, only some observational studies have pointed to an association between platelet inhibition by acetylsalicylic acid and improved outcomes in the clinical setting [4, 5] while more recent data could not confirm these findings [6]. In addition to platelet activation, platelet turnover is increased in sepsis and levels of young, immature platelets may be associated with worse outcomes in this setting [7]. In cardiovascular patients, immature platelets have been identified as the strongest predictor for a reduced effectiveness of thienopyridines in stable patients undergoing percutaneous coronary intervention [8]. Recently, our group reported that immature platelets have a set of unique properties as compared to older platelets [9]. Also, transcriptome degradation and enrichment of circular RNAs in platelets [10] may be reduced in immature platelets. Thus, sepsis as a state of increased platelet turnover may be seen as a model to study changes in platelet properties associated with higher platelet turnover.

Indeed, it was recently reported that human and mouse platelets display an altered transcriptome in sepsis [11]. Significant correlations were found between changes in human and mouse platelets, with 542 shared transcripts concordantly regulated in both species [11]. Upregulated transcripts were functionally associated with membrane, cell-cell adhesion junction, focal adhesion, and cytoplasm activities [11]. Yet, it was not reported whether the platelet turnover was increased while mean platelet volume as a surrogate marker of platelet production was similar between septic patients and healthy controls [11].

In the present study, we simultaneously assessed platelet turnover by measurement of the immature platelet fraction as well as the platelet transcriptome in patients with sepsis compared to stable control patients. Thus, we sought to determine whether changes in the platelet transcriptome are associated with an increased platelet turnover, aiming to shed light on specific characteristics of immature platelets.

## Material and methods

### Subjects, inclusion and exclusion criteria, blood sampling

Blood samples were collected from 8 patients with sepsis requiring vasopressor therapy and 8 control patients between September 2015 and February 2016. Inclusion criteria for sepsis were a CRP-level > 5 mg/dl in combination with a clinically manifest infection, a procalcitonin level > 0.05 ng/ml, or positive blood cultures. Patients or legally authorised representatives were approached for participation on day 1 or 2 after admission to the intensive care unit.

Control patients all had stable coronary artery disease and presented similar biometrical and demographic characteristics as septic subjects, though lacking acute or chronic signs of infection as well as signs of acute illness. Patients were approached for participation when admitted for elective coronary angiography. Blood sampling was performed before coronary angiography.

Exclusion criteria in all patients were patients receiving platelet or clotting factor transfusions, patients with known platelet disorders or with heparin induced thrombocytopenia, patients undergoing acute or chronic haemodialysis or peritoneal dialysis, and patients with a malignancy or a haematological disorder.

The entire patient cohort represents a male cohort with cardiovascular and coronary artery disease. The patient cohort may not be representative for a larger cohort of patients with sepsis, since peritonitis and abdominal sepsis were not present in the cohort.

Blood samples were drawn by arm vein puncture with a safety Multifly-Set® (Sarstedt, Nümbrecht, Germany) and collected in two citrate monovettes (10 ml, 1 ml trisodium-citrate, Sarstedt, Nümbrecht, Germany), one EDTA monovette (2.7 ml, 1.6 mg EDTA, Sarstedt, Nümbrecht, Germany), one Hirudin monovette (2.7 ml, > 525 antithrombin units Hirudin/ml, Sarstedt, Nümbrecht, Germany) and processed within 15 minutes.

Each patient or a legally authorised representative provided written informed consent to participate in the study which was approved by the Ethics Committee of the Albert-Ludwigs-University Freiburg (EK 24/15). The investigation conformed to the principles outlined in the Declaration of Helsinki.

### Impedance aggregometry

Hirudin monovettes were used for impedance aggregometry to test platelet reactivity with the ASPItest, the ADPtest and the TRAPtest according to the manufacturer's recommendations (Multiplate®, Roche Diagnostics, Basel, Switzerland): recording the area under the curve of aggregation units over 6 minutes. Results are reported as aggregation units $^*$ minute (AU $^*$ min).

### Immature platelet fraction (IPF) and immature platelet count (IPC)

EDTA samples were used to determine immature platelet fraction (IPF), immature platelet count (IPC), and total platelet count using a Sysmex XE 2100 device (Sysmex Corporation, Kobe, Japan).

### Purification of platelets by leukocyte depletion and RNA isolation

Citrate monovettes were used to extract platelet-rich, leukocyte-depleted plasma (LD-PRP). The tubes were centrifuged at 2,040 U/Min. (750 g) for 2 minutes (Megafuge 1.0 RS, Haereus, Hanau, Germany). The supernatant (platelet-rich plasma, PRP) was carefully pipetted into a 50 ml FALCON tube (Becton Dickinson, New Jersey, United States). Platelet concentration in the PRP was analysed by an ABX–Micros 60 (HORIBA Ltd, Minami-ku Kyoto, Japan).

Leucocyte depletion was achieved through negative bead separation by application of the EasySep® Human Whole Blood CD45 Depletion kit according to the manufacturer's recommendations (StemCell Technologies Inc., Vancouver, Canada).

The leukocyte-depleted platelet pellet was resuspended in 750 μl QIAzol Lysis Reagent (Qiagen GmbH, Hilden, Germany). The suspension was vortexed for 1 minute and immediately stored at -80˚C. RNA was extracted from the platelet-QIAzol Lysis Reagent aliquots by application of the RNeasy Mini kit (Qiagen GmbH, Hilden, Germany) according to the manufacturer's recommendations. Extracted RNA was then immediately stored at -80˚C.

## RNA library preparation and sequencing

RNA yield was determined by Qubit RNA HS assay kit (ThermoFisher Scientific, Reinach, Switzerland). 50 ng of total RNA were subjected to depletion of ribosomal RNA by the use of the RiboMinus™ Eukaryote System v2 kit (ThermoFisher Scientific/Life technologies™/ Ambion®, Reinach, Switzerland). As next step, RNA library preparation was achieved by application of the NEXTflex™ Illumina RNA-Seq Library Prep Kit v2 with Molecular Indexes by Bioo Scientific© (Bioo Scientific, Austin, Texas, United States) according to the manufacturer's recommendations. Library size was assessed before sequencing using the Agilent 2100 Bioanalyzer high sensitivity DNA analysis kit (Agilent, Waldbronn, Germany). Barcoded libraries were sequenced on an Illumina HiSeq 2500 in 75bp paired-end high output mode.

## Statistical analysis

Discrete variables are reported as counts (percentages) and continuous variables as median with interquartile range. For discrete variables, we tested differences between groups with the $\chi$2-test or Fisher's exact test when expected cell sizes were less than 5. To compare continuous variables, the Mann-Whitney-U test was used. All tests were two-sided and results were regarded as statistically significant at a $\alpha$-level 5%. IBM SPSS statistics, version 23.0 (IBM corporation, Armonk, NY, USA) was used for statistical analysis of clinical data.

## RNA-sequencing data analysis

Bioinformatic data analysis was performed mainly (quality control, alignment, and quantification of expression levels) on the Freiburg Galaxy platform (now located at usegalaxy.eu). Reads were mapped to hg38 genome with the HISAT aligner [12]. Exploratory data analysis was performed with pcaExplorer, an R/Bioconductor package (v2.16.0) [13], with Principal Component Analysis to investigate the association with various clinical parameters as well as the immature platelet fraction, using as input the normalised transformed expression values. In detail, pcaExplorer was used with either the top 5,000 or the top 500 expressed genes as input. Directional RNA library duplicate removal was performed with Je-MarkDuplicates [14], followed by estimation of differential gene expression by use of DESeq2 [15], in the framework provided by the ideal package (v1.14.0) [16]. Genes were considered differentially expressed if p-adj (p value corrected by Benjamini-Hochberg procedure) was < 0.05 from DESeq2 and log2 fold change was > ± 1.5. According to RNASeqPower—a Bioconductor /R package for sample size calculation in RNASeq studies [17]—with 8 patients, an $\alpha$ of 0.05, effect sizes we desire to detect of > 2.83 (log2 fold change > 1.5), and coefficient of variation of 0.6, we expect to achieve a power of 0.9. The use of DESeq2, the tool used for assessment of differential gene expression, was considered adequate in this number of samples [18]. Gene cluster analysis and data visualization were performed with ClueGO and Integrative Genomics Viewer (IGV), respectively [19, 20]. Transcripts visualised in IGV were normalised by Fragments Per Kilobase of transcript, per Million mapped reads (FPKM). In ClueGO, separate

analyses were performed using only GO terms and using Reactome [21] and KEGG [22] pathways together with GO terms. For evaluation of activating and inhibiting pathways in platelets [23] (S1 Table), Gene Set Enrichment Analysis (GSEA) [24] was performed. GSEA was also carried out with previously published genesets of genes regulated in platelets from patients with sepsis [11] or COVID-19 [25] as well as from reticulated platelets [9]. To recapitulate previously published, differential gene expression in platelet from patients with sepsis, RNA-Seq data was downloaded from SRA (PRJNA521077) and re-analysed as described above. PTESfinder, a computational method for identification of post-transcriptional exon shuffling (PTES) events was used as described [26] in a Linux environment to detect the presence of circular RNAs in the RNA-Seq data. PTES were analysed for differential expression using edgeR [27].

## Results

### Baseline and laboratory characteristics

Considering demographic characteristics, septic (n = 8) and control patients (n = 8) were all male and were well matched (Table 1). As expected, significant differences were seen in key laboratory values such as C-reactive protein (21.1 [15.2–28.9] mg/dl in sepsis vs. 0.2 [0.0–0.8] in controls) and levels of creatinine clearance (19 [13–44] ml/min/1.73m$^2$ in sepsis vs. 62 [41–88] ml/min/1.73m$^2$ in controls). Further significant differences were seen in levels of haemoglobin (10.0 [9.1–10.2] g/dl in sepsis vs. 12.9 [12.2–14.9] g/dl in controls, p = 0.001) and LDL-

**Table 1. Patient characteristics.**

| | Entire cohort (n = 16) | Control (n = 8) | Sepsis (n = 8) | p value (sepsis vs. control) |
|---|---|---|---|---|
| **Clinical parameters** | | | | |
| Age (years) | 75 [69–78] | 76 [72–80] | 72 [62–77] | 0.127 |
| Female sex | 0 (0%) | 0 (0%) | 0 (0%) | 1 |
| Body mass index (kg/m$^2$) | 29 [27–34] | 29 [28–34] | 28 [25–32] | 0.328 |
| Diabetes mellitus | 6 (38%) | 2 (25%) | 4 (50%) | 0.608 |
| Coronary artery disease | 13 (81%) | 8 (100%) | 5 (63%) | 0.200 |
| Reduced LV function | 8 (50%) | 4 (50%) | 4 (50%) | 1 |
| Peripheral artery disease | 4 (25%) | 2 (25%) | 2 (25%) | 1 |
| **Laboratory characteristics** | | | | |
| Haemoglobin (g/dl) | 11.8 [10.0–12.9] | 12.9 [12.2–14.9] | 10.0 [9.1–10.2] | 0.001 |
| Leukocytes (10$^3$/μl) | 8.8 [7.1–12.8] | 7.4 [6.8–9.0] | 10.9 [7.7–15.5] | 0.093 |
| C-reactive protein (mg/dl) | 7.7 [0.2–21.9] | 0.2 [0.0–0.8] | 21.1 [15.2–28.9] | 0.001 |
| Creatinine-Clearance (ml/min/1.73m$^2$) | 42 [15–64] | 62 [41–88] | 19 [13–44] | 0.016 |
| LDL cholesterol (mg/dl) | 93 [40–127] | 115 [86–151] | 49 [14–98] | 0.027 |
| HDL cholesterol (mg/dl) | 43 [35–50] | 45 [43–51] | 35 [11–48] | 0.058 |
| Triglycerides (mg/dl) | 135 [108–184] | 163 [118–193] | 109 [80–155] | 0.083 |
| **Medication** | | | | |
| Acetylsalicylic acid | 11 (69%) | 6 (75%) | 5 (63%) | 1 |
| P2Y12 inhibitors | 7 (44%) | 5 (63%) | 2 (25%) | 0.315 |
| Oral anticoagulation | 3 (19%) | 0 (0%) | 3 (38%) | 0.200 |
| Heparin | 7 (44%) | 2 (25%) | 5 (63%) | 0.315 |
| Statins | 10 (63%) | 7 (87%) | 3 (38%) | 0.119 |

LV = left ventricular, LDL = low-density lipoprotein, HDL = high-density lipoprotein, P2Y12: Purinergic Receptor P2RY12. Fisher's exact test for ordinal/categorical variables, Mann-Whitney U test for continuous variables, p values < 0.05 regarded as significant.

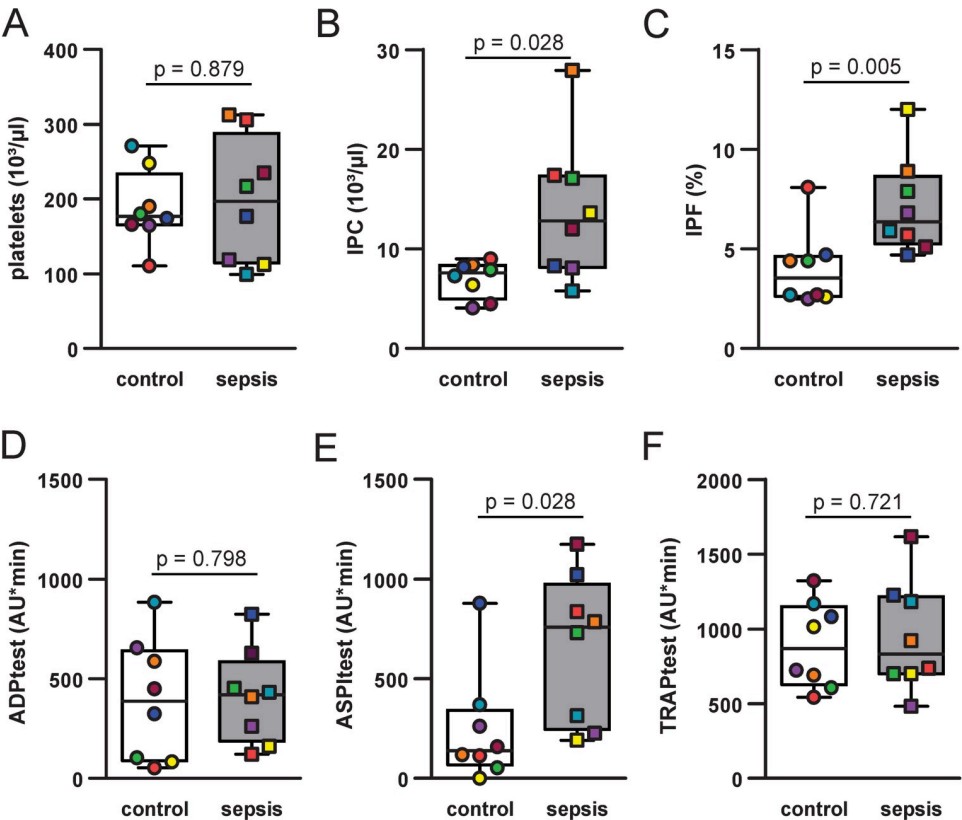

**Fig 1. Platelet parameters and platelet reactivity.** Comparison between control patients (transparent bars) and patients with sepsis (filled bars) in A-F. Platelet counts (A), immature platelet count [IPF] (B) and immature platelet fraction [IPF] (C) in upper row. Multiplate aggregometry with stimulation by adenosine diphosphate at a final concentration of 6.5μM [ADPtest] (D), by arachidonic acid at a final concentration of 0.5 mM [ASPItest] (E) and by thrombin-receptor associated protein 6 at a final concentration of 32 μM [TRAPtest] (F) in lower row. Filling colour of data points indicates individual samples within the respective groups, boxes with inner line display interquartile range with median, whiskers minimum and maximum of values. P values from Mann-Whitney U test.

cholesterol (49 [14–98] mg/dl in sepsis vs. 115 [86–151] mg/dl in controls, p = 0.027) (Table 1).

## Immature platelet levels and platelet function tests

Septic patients presented with an increased IPF (7.1 [4.7–12] % in sepsis vs. 4.0 [2.5–8.1] % in controls) and IPC (13.8 [5.8–27.9] $10^3$/μl in sepsis vs. 7.0 [4.1–9.0] $10^3$/μl in controls). Multiplate electric impedance aggregometry showed no differences on ADP and TRAP simulation between the two groups. Stimulation with arachidonic acid showed an increased platelet aggregation in septic patients compared to control (661 [193–1175] AU*min in sepsis vs. 245 [0–880] AU*min in controls, p < 0.05) (Fig 1).

## Analysis of post-transcriptional exon shuffling (PTES) transcripts

Platelets have been described as cells with a degraded transcriptome which is enriched for circular RNA [10]. Given the higher platelet turnover in patients with sepsis, as evidenced by elevated IPF levels, we were interested if we could detect a decrease of putative circular RNAs in platelets from septic patients. Using PTESfinder [26], PTES transcripts were identified and normalised to the read number per sample (Fig 2A) or to respective gene expression (Fig 2B).

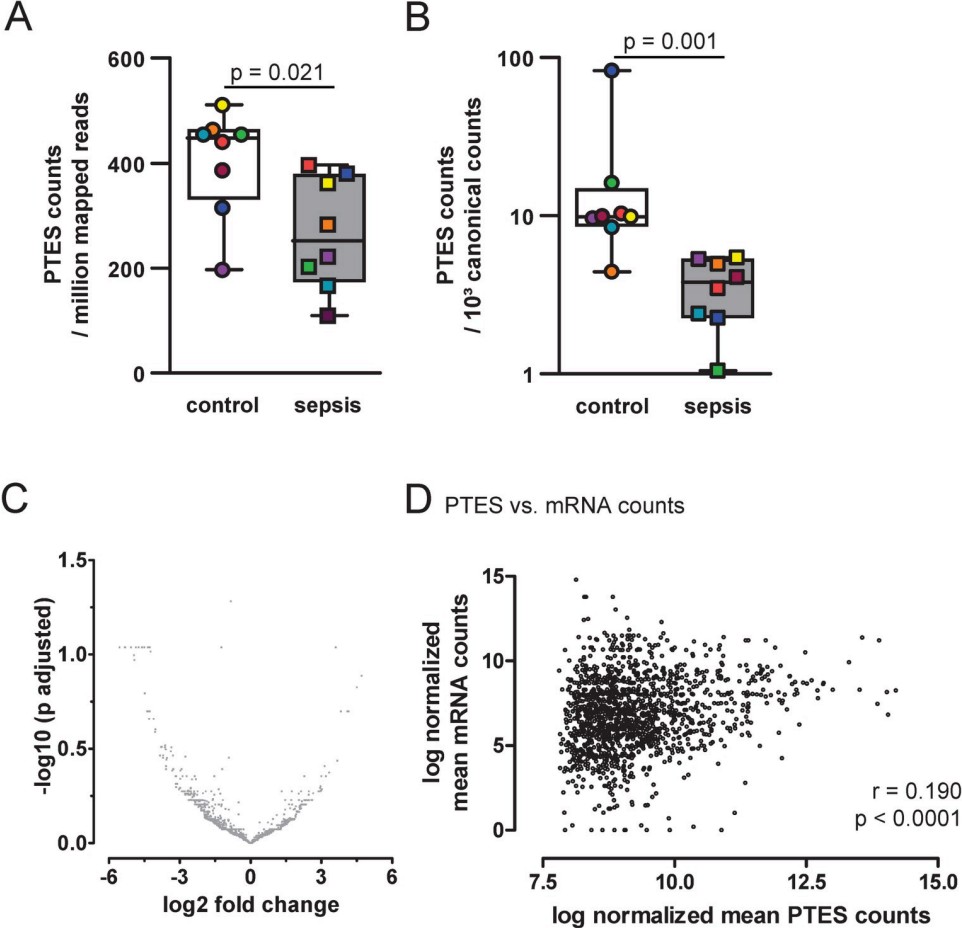

**Fig 2. Analysis of post-transcriptional exon shuffling (PTES) transcripts.** Comparison between control patients (transparent bars) and patients with sepsis (filled bars) in A and B. PTES transcripts or putative circular RNA transcripts were identified and normalised for each patient to (A) the overall number of mapped reads or (B) to the sum of canonical reads as identified by PTESfinder. Filling colour of data points indicates individual samples within the respective groups, boxes with inner line display interquartile range with median, whiskers minimum and maximum of values. P values from Mann-Whitney U test (C) Volcano plot of adjusted p values from differential expression analysis by edgeR versus log2 fold changes showing absence of significant differences between control patients and patients with sepsis. Note the different y-axis as compared to Fig 5. (D) Spearman correlation between mean mRNA expression and mean PTES expression levels. Note the off-set x-axis, limiting PTES counts to values > 8. Lower values were filtered out for low expression in differential expression analysis.

Both ways of normalisation methods showed a significant decrease of putative circular RNA in platelets from septic patients. Further evaluation did not show a differential regulation of PTES transcripts in platelets from septic patients (Fig 2C) but a low, yet highly significant correlation between mean PTES and respective linear transcript expression (Fig 2D).

## Global transcript distribution and purity of ribosome-depleted platelet RNA-sequencing

Duplicate removal by Je [14] identified a large proportion of PCR duplicates among mapped reads—ranging from 49.9% to 96.4% (S1 Fig). In both groups, most expressed transcripts were protein-coding genes and only minor changes occurred in the number of consistently expressed genes, defined as expression present in all samples and a median normalised count

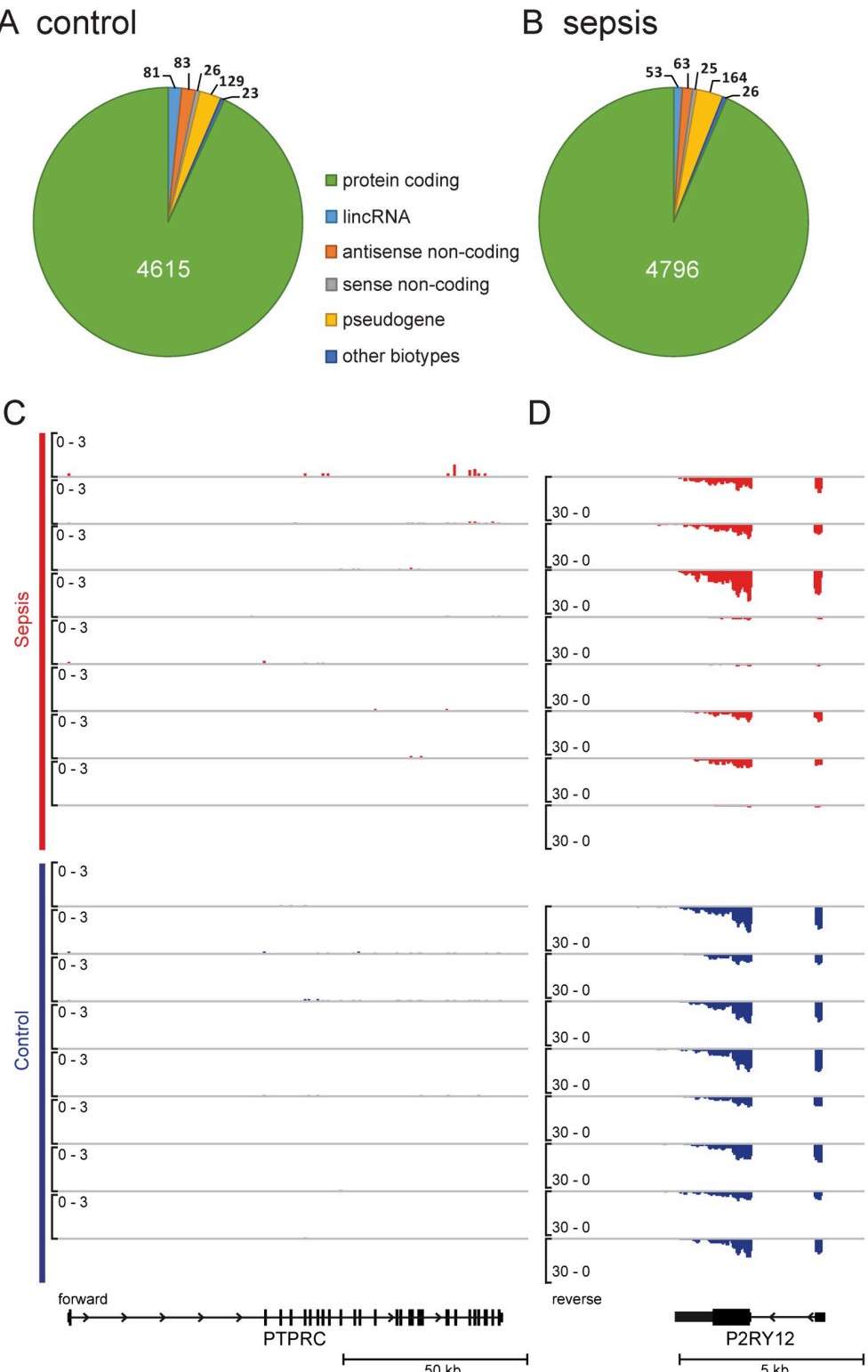

**Fig 3. Global transcript expression and purity of platelet RNA-Seq.** Illustration of global transcript expression in platelets and purity of platelet RNA-Seq separately for each group. A and B show consistently expressed transcripts, defined as a median normalised count of $\geq$ 10 and expression across all 8 samples of each cohort, in control patients (A) and patients with sepsis (B). Genome Browser views of Protein Tyrosine Phosphatase Receptor Type C [PTPRC] gene / CD45 gene (C) and Purinergic Receptor P2RY12 [P2RY12] (D) expression. Individual samples from patients

with sepsis are shown in the top part (indicated by left-sided red bar), individual samples from control patients are shown in the lower part (indicated by left-sided blue bar). Scaling in black brackets indicates fragments per kilobase per million mapped reads (FPKM). lincRNA = long intergenic non-coding RNA, kb = kilo base pair.

of ≥ 10 (Fig 3A and 3B and S2 Table). Also, patterns of transcript distribution were largely similar across all samples when a normalised count ≥ 10 was used as cut-off for transcript expression (S2 Fig). Regarding a potential leukocyte contamination of the platelet RNA-Seq, median CD45 expression measured by the ENSG00000081237 gene was 3.0 [1.3–6.7] normalised counts in samples from patients with sepsis and 4.6 [2.0–7.5] normalised counts in samples from control patients (p = 0.57). Single views for each sample demonstrate that there was no consistent presence of CD45 exonic reads (Fig 3C). Conversely, a consistent number of reads throughout the entire coding sequence of the P2Y12 receptor gene was detected (Fig 3D).

## Clinical parameters associated with sepsis and global platelet gene expression

To evaluate the interplay between clinical variables that differ between sepsis and control patients and global gene expression, a PCA model illustrating the values of these variables by colour coding was used. Principal component analysis was performed on the top 5000 expressed genes (Fig 4A, 4C and 4E), essentially falling together with consistently expressed genes as well as for the top 500 expressed genes (Fig 4B, 4D and 4F). As clinical variables, IPF, C-reactive protein, and haemoglobin showed highly significant differences between the two groups, especially in samples sepsis_2 and sepsis_3 as compared to control samples. Conversely, these samples clustered closely together with control samples in the PCA model.

## Differentially expressed genes in platelets from patients with sepsis

As next step, we assessed differential gene expression between the two groups. No adjustment for clinical characteristics with significant differences was performed since all these variables are directly affected by sepsis. Profound changes in gene expression were detected in samples from patients with sepsis. Applying similar strict criteria as before [11], 524 genes were identified as upregulated in sepsis, whereas 118 genes were regarded as downregulated (Fig 5A and S2 Table). If a similar analysis with no PCR duplicate removal was performed, only 68 transcripts would have been considered as upregulated and 55 transcripts as downregulated (data not shown). An expression heatmap of the differentially expressed genes revealed inter-individual heterogeneity in gene expression (Fig 5B). For instance, in the samples Control_2, Sepsis_2 and Sepsis_3, a considerable part of the upregulated genes showed expression values fitting to the opposite clinical group (Fig 5B, upper part of heatmap). If sample clustering were performed, sample Control_2 would cluster with the sepsis samples, despite the restriction to differentially expressed genes (data not shown). Among genes with consistent regulation across all samples were interferon-induced transmembrane proteins 2 and 3 (*IFITM2*, *IFITM3)* or small GTP binding protein Rac2 (*RAC2)*, being upregulated (Fig 5B and 5C). MAP3K7 C-Terminal Like (*MAP3K7CL)*, long intergenic non-coding RNA 000346 (*LINC00346*) or Lysophospholipase-like 1 (*LYPLAL1*) are examples of consistently downregulated genes (Fig 5B and 5D).

## Functional annotation of differentially expressed genes

To gain further insight into possible functional implications of differential gene expression, we performed separate unrestricted gene ontology analyses of genes that were up- or

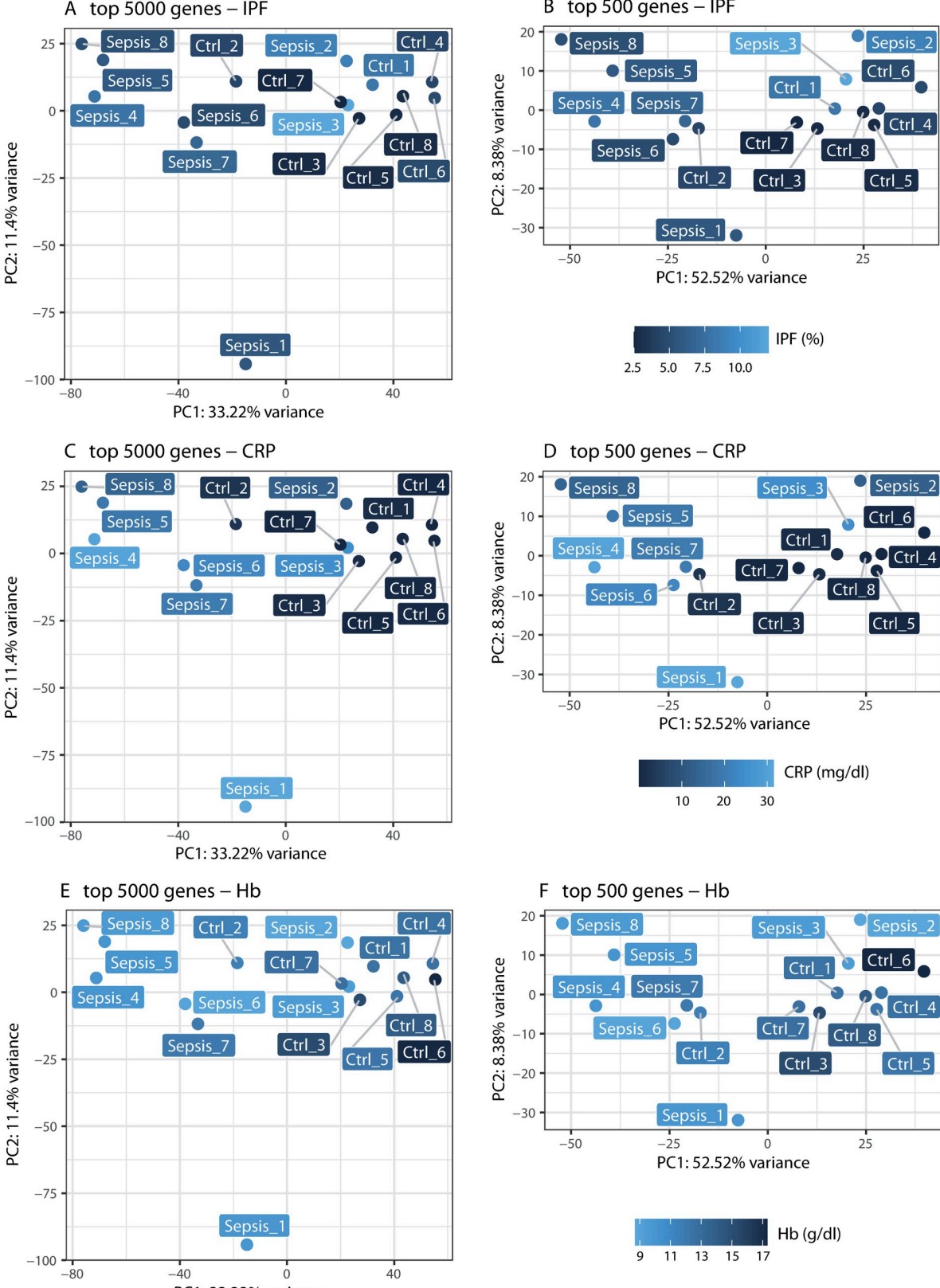

**Fig 4. Principal component analysis of top 5000 and top 500 expressed genes.** Principal component analysis of top 5000 expressed (A, C, E) and top 500 expressed genes (B, D, F). Values of clinical parameters with most significant differences between groups are displayed by respective shades of blue, according to color gradation bars. IPF = immature platelet fraction, CRP = C-reactive protein, Hb = hemoglobin, PC = principal component.

downregulated in platelets from septic patients using ClueGO [19]. Applying similar computational parameters, upregulated genes were identified as enriched in 29 biological process GO terms (Fig 6A) whereas downregulated genes were regarded as enriched in 1 molecular function and 8 biological process GO terms (Fig 6B). GO terms pertaining to upregulated genes were mostly related to catabolic processes and protein translation. 27 of the 29 GO terms formed a closely interlinked network describing the entire translational process from amide biosynthesis to sub-cellular protein localization (Fig 6A, lower part). Conversely, GO terms of processes enriched for downregulated genes were less connected. 4 of the 9 GO terms formed one network associated with the regulation of myeloid cell differentiation, while the remaining five terms were not connected to any other term. When adding Reactome [21] and KEGG [22] pathways to the analysis, many pathway terms were identified as enriched within upregulated genes. These were predominantly related to translational activity but also to inflammatory processes (S3A Fig). Regarding the downregulated genes, only few pathway terms were enriched (S3B Fig).

## Analysis of prespecified gene sets of platelet activation signaling

We also sought to evaluate if *a priori* defined gene sets were enriched in platelets from septic patients. To this end, we performed GSEA [24] with published gene sets of signaling pathways that were either related to platelet activation or were involved in inhibiting platelet activation [23] (S1 Table). Two of the 29 gene sets, BIOCARTA_EPHA4_pathway and KEGG_Calcium_-signaling_pathway—both related to platelet activation—showed a nominal p value < 0.05 and normalised enrichment scores > 1.3 (Fig 7, upper part). The normalised enrichment score (NES) represents the primary metric since it allows comparison between gene sets by taking the gene set size and correlations between the expression data and the gene sets into account [24]. In both gene sets, genes with core enrichment showed higher expression in platelets from control patients (Fig 7, lower part). Core enrichment comprises the leading edge subset of genes that add most to the enrichment score, being ranked at the top of the gene set before a positive peak enrichment score [24].

## Comparison of current transcriptome to differential gene expression in platelets from other patients with sepsis, patients with COVID-19, and reticulated platelets

We also used GSEA to compare the current changes in gene expression with recent platelet RNA-Seq analyses from patients with sepsis [11], COVID-19 [25] and from reticulated platelets [9]. As expected, gene sets derived from a cohort of 5 patients with sepsis and 5 healthy donors showed highly significant enrichment in our data set (Fig 8A and 8B). A similar observation was made with genes regulated in platelets from patients with COVID-19 [25] (Fig 8C and 8D). Of note, no enrichment of genes reported to be upregulated in reticulated platelets [9] was observed in platelets from septic patients (Fig 8E). Conversely, concerning genes downregulated in reticulated platelets as compared to platelets with low RNA content [9], we observed an enrichment of this gene set in platelets from control patients (Fig 8F).

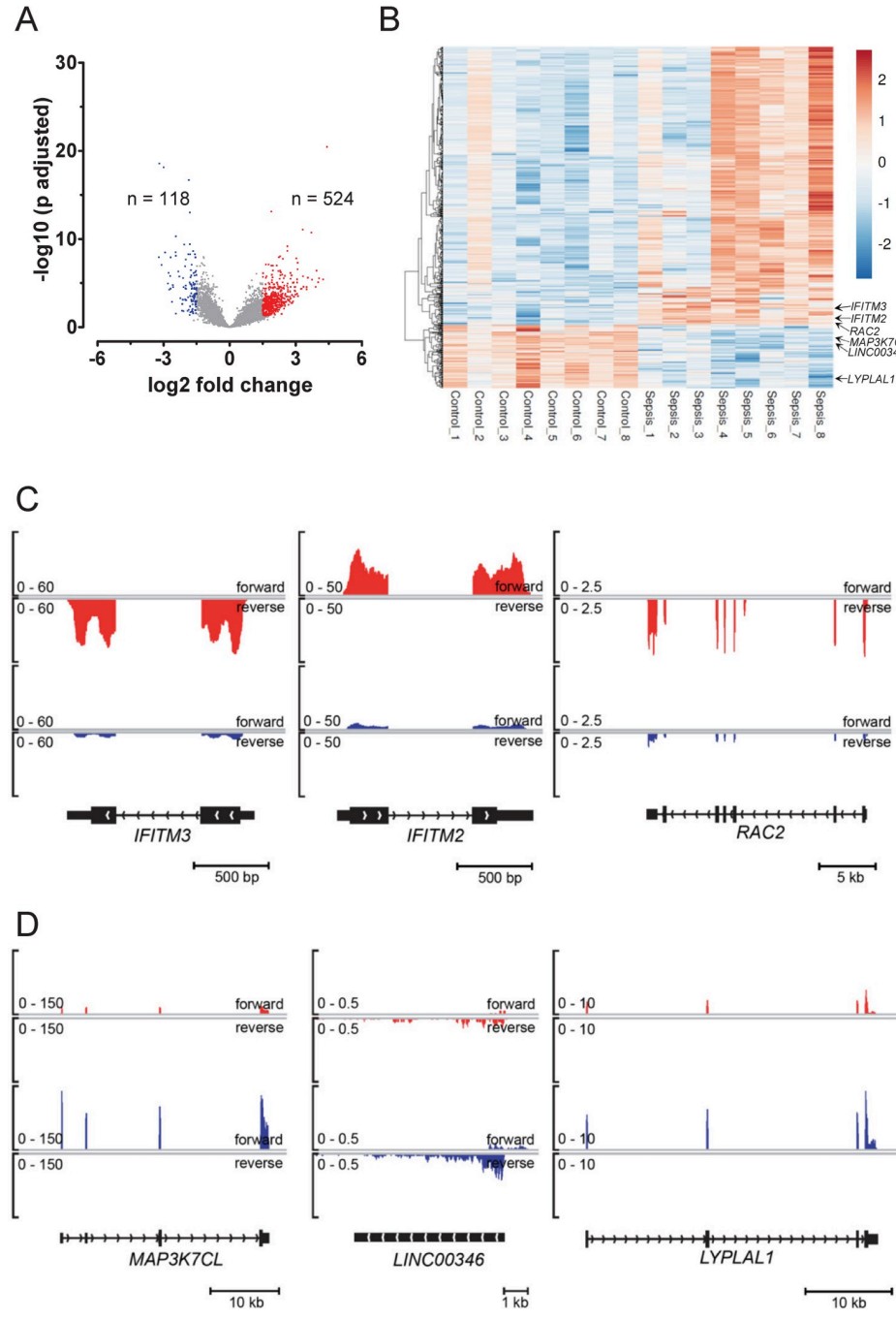

**Fig 5. Differential gene expression in platelets from patients with sepsis.** (A) Volcano plot of differential gene expression analysis by DESeq2. Blue dots indicate 118 downregulated genes; red dots 524 upregulated genes. Dots in light gray represent genes without significant regulation. (B) Differential gene expression illustrated by a heatmap with transcript clustering. Normalized gene expression is represented between dark red for higher expression and dark blue for lower expression than the average. Position of representative consistently expressed genes are indicated by arrows. (C) Genome Browser views of representative, upregulated genes *IFITM2*, *IFITM3* and *RAC2*. (D) Genome Browser views of representative, downregulated genes *MAP3K7CL*, *LINC000346* and *LYPLAL1*. Scaling in black brackets indicates fragments per kilobase per million mapped reads (FPKM); merged expression of all samples in each group is shown. lincRNA = long intergenic non-coding RNA, bp = base pair, kb = kilo base pair.

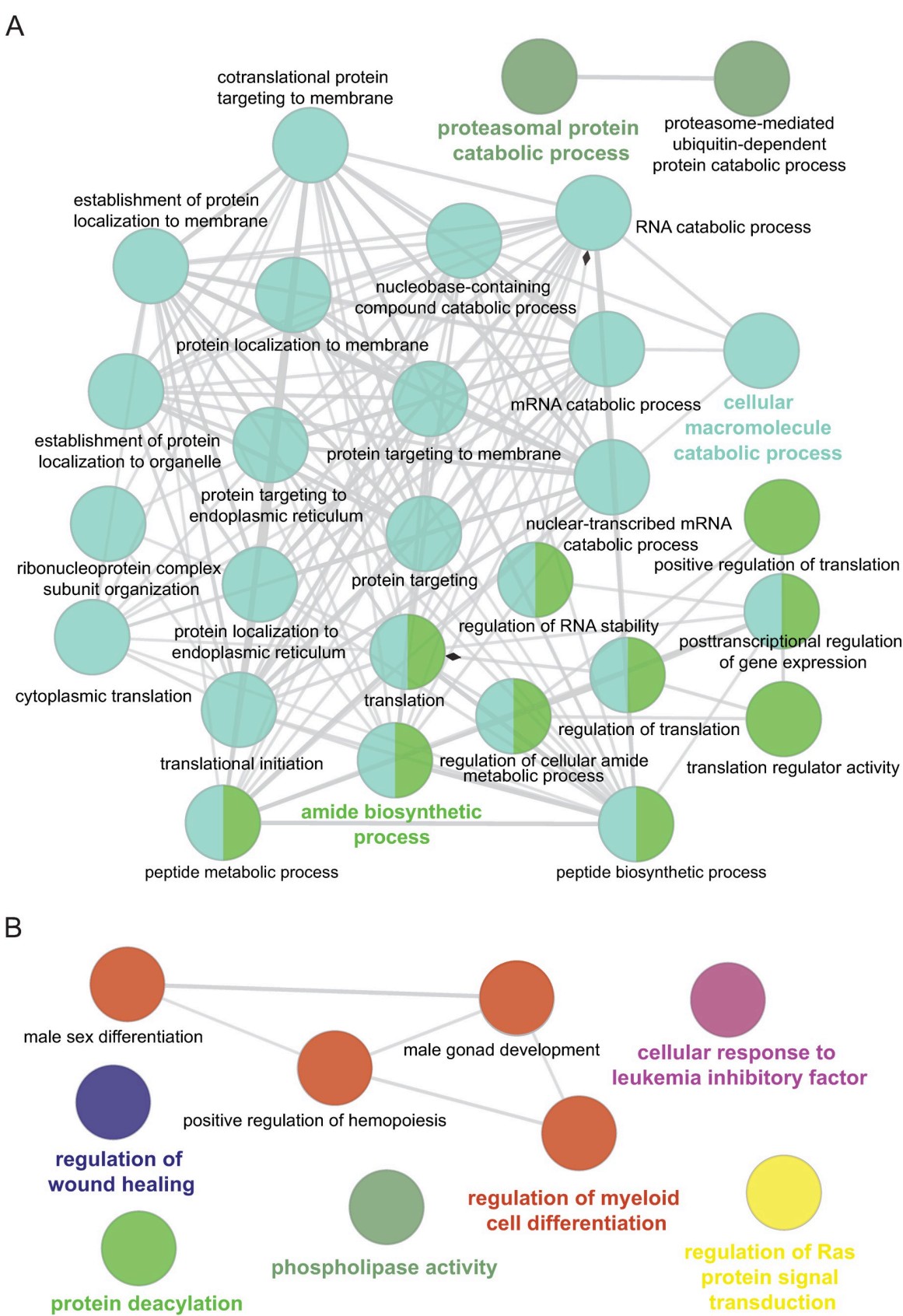

**Fig 6. Functional annotation of genes differentially expressed in platelets from septic patients versus platelets from control patients.** Functional annotation of differentially expressed genes is analysed by ClueGO, a Cytoscape plugin, for upregulated genes (A) and downregulated genes (B) with similar parameters using Gene Ontology (GO) terms. Terms with highest significance within one network are spelled in the respective colour.

## Discussion

In this study, we provide a detailed analysis of alterations in the platelet transcriptome in sepsis patients. While results from RNA-sequencing in platelets from septic patients have been reported recently [11], this is to our knowledge the first study to simultaneously analyse levels of immature platelets in sepsis. Also, by magnetic bead-based leukocyte depletion, we obtained highly purified RNA for RNA-sequencing as previously described [28]. For the first time, we introduced the use of unique molecular identifiers to eliminate PCR amplification biases which may be relevant in a transcriptome with reduced complexity as it is observed in platelets [10].

This study has four main findings: (i) in patients with sepsis, levels of immature platelets were significantly elevated; (ii) putative circular RNA transcripts were decreased in platelets from patients with sepsis; (iii) profound changes in protein-coding transcripts pointing to increased protein translation were present in platelets from patients with sepsis and (iv) transcripts upregulated in reticulated platelets from healthy donors were not enriched in platelets from patients with sepsis.

Several studies have demonstrated increased levels of immature platelets in patients with sepsis [7]. The present data demonstrate for the first time that putative circular RNAs are decreased in platelets from septic patients, pointing to a less degraded transcriptome associated with increased levels of immature platelets. However, the biological implications of the increased turnover remain unclear. Phenotypic studies have not led to conclusive evidence that platelets from septic patients are a priori hyperreactive due to the increased levels of reticulated platelets [29–31]. Recent data from Middleton showed an upregulation of ITGA2B and increased platelet reactivity by increased PAC-1 binding in a large cohort, yet reticulated platelets were not quantified [11]. Instead, mean platelet volume, as a surrogate marker of platelet turnover [32], was found to be similar between patients with sepsis and controls. In the present study, we show that immature platelet levels are elevated in patients with sepsis and that transcriptomic changes, such as the closely interlinked network of functional annotations describing the entire translational process, strongly point to increased protein translation in megakaryocytes or platelets. In this context, it should however be noted that we neither evidenced active translation in platelets nor did we observe an association between IPF levels and global gene expression. Rather, clinical parameters with relevant, significant differences between sepsis and control patients did not seem to match with levels of selected transcripts as illustrated by principal component analysis. Also, in GSEA analyses, we observed a high congruence with gene expression changes observed in another sepsis cohort or in patients with COVID-19 while no enrichment of genes upregulated in reticulated platelets from healthy volunteers was seen.

In contrast to genes upregulated in sepsis, we observed that genes with higher expression in platelets with low RNA content were enriched in platelets from control patients. Thus, it could be hypothesized that linear RNAs with low tendency to degradation become more abundant in older platelets as it has been shown for degradation-resilient circular RNAs [10].

Concerning platelet function as assessed by impedance aggregometry, platelets from septic patients showed a significantly increased aggregation in the ASPItest. At first sight, our findings appear to stand in contrast with previous studies [33–35] which showed a reduction of

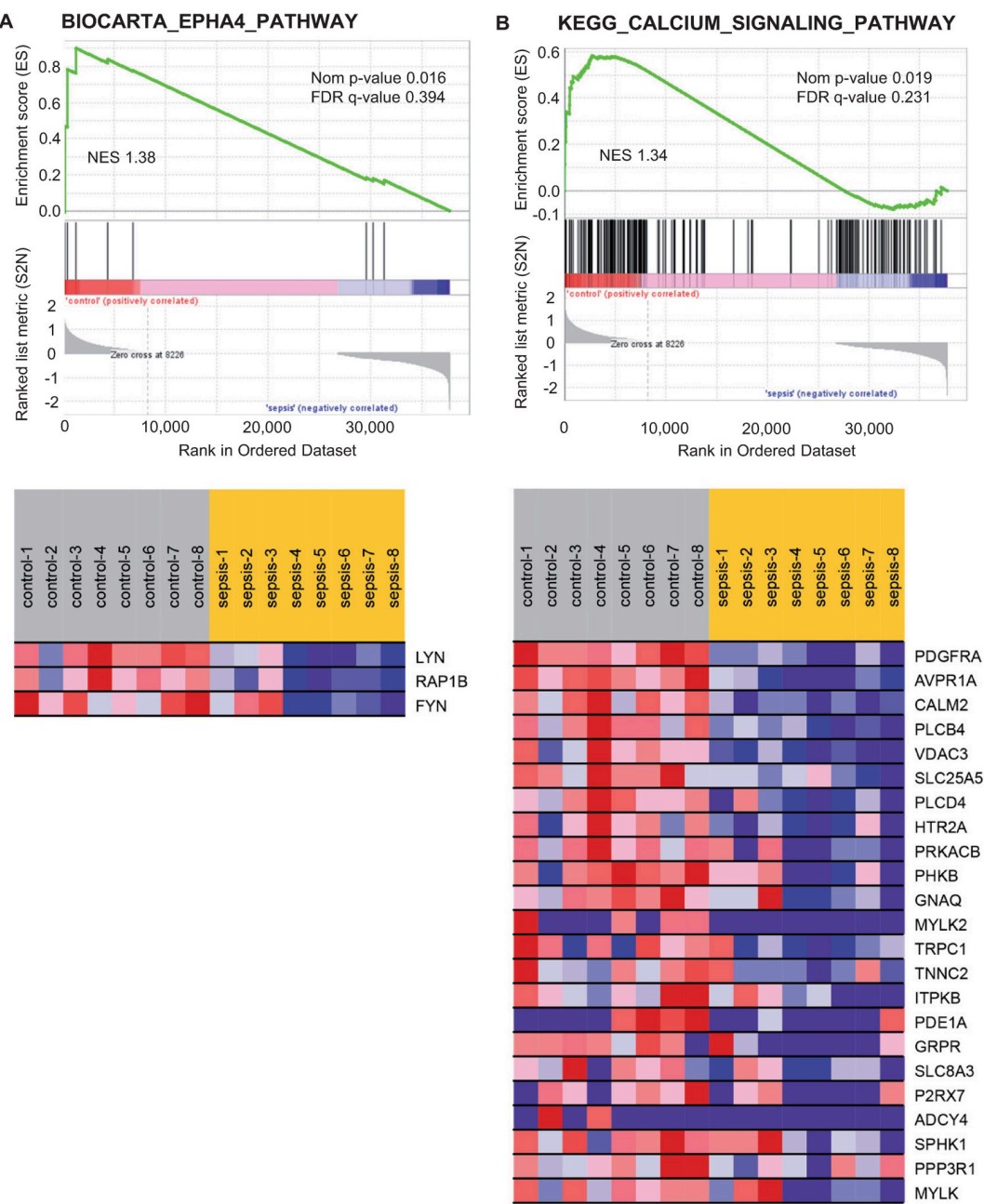

**Fig 7. Gene Set Enrichment Analysis with predefined gene sets related to platelet activation identifies two gene sets enriched in the platelet transcriptome from patients with sepsis compared to control patients.** Shown are results for BIOCARTA_EPHA4_PATHWAY (A) and KEGG_CALCIUM_SIGNALLING_PATHWAY (B), two of 29 analysed gene sets (S1 Table) enriched in platelets from patients with sepsis, at a nominal (Nom) p value of < 0.05 and normalised enrichment scores (NES) of > 1.3. The normalised enrichment score represents the primary metric since it allows comparison between gene sets by taking the gene set size and correlations between the expression data and the gene sets into account [24]. Genes within the core enrichment (dark red in ranked list metric) are depicted with respective expression values as heatmap below. Core enrichment comprises the leading edge subset of genes that add most to the enrichment score, being ranked at the top of the gene set before a positive peak enrichment score (maximal value of green line) [24].

platelet aggregation in patient with septic shock. However, it must be noted that in these studies, patients were apparently not treated with acetylsalicylic acid. In our study, medication with acetylsalicylic acid was common and balanced between the two groups. Thus, the

A  ALTERNATE SEPSIS_UPREGULATED

B  ALTERNATE SEPSIS_DOWNREGULATED

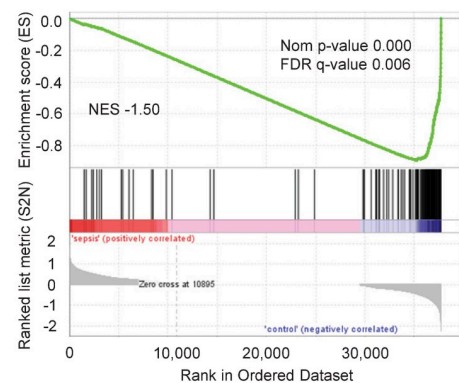

C  COVID19_UPREGULATED

D  COVID19_DOWNREGULATED

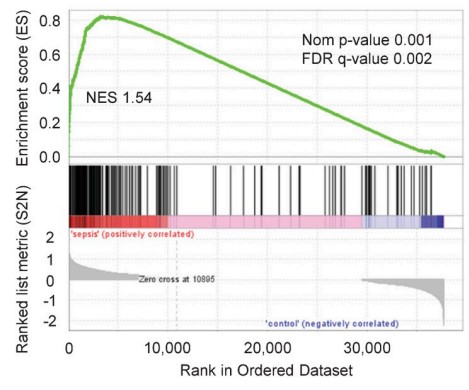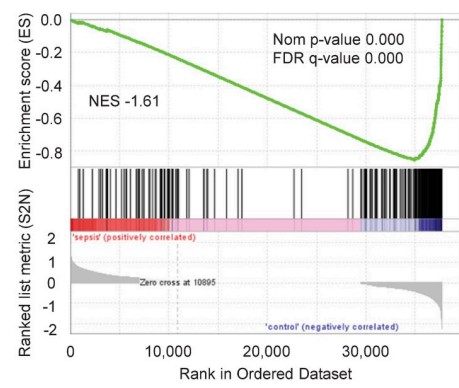

E  RETICULATED PLATELETS_UPREGULATED

F  RETICULATED PLATELETS_DOWNREGULATED

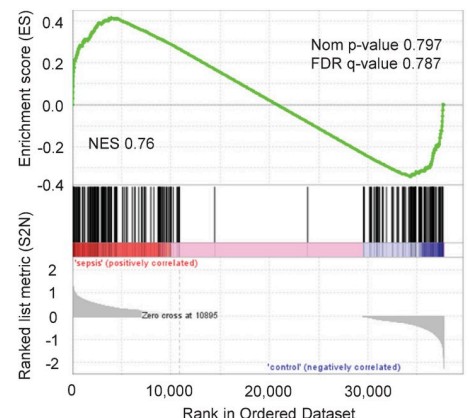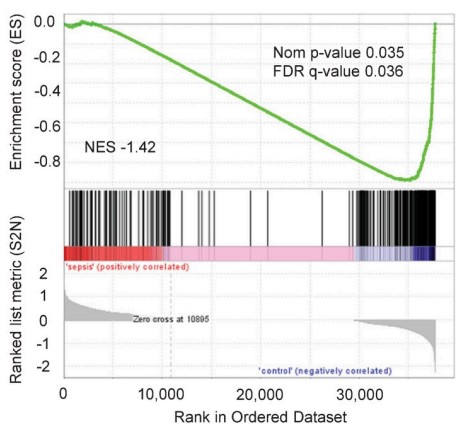

**Fig 8. Gene expression in platelets from patients with sepsis and control patients is compared with platelet transcriptome changes in other patients with sepsis, patients with COVID-19 and reticulated platelets by Gene Set Enrichment Analysis.** Gene Set Enrichment Analysis with gene sets of differentially expressed genes from published platelet RNA-Seq data [9, 11, 25]. (A) Genes upregulated in sepsis, (B) genes downregulated in sepsis. (C) Genes upregulated in COVID-19, (D) genes downregulated in COVID-19. (E) Genes upregulated in reticulated platelets, (F) genes downregulated in reticulated platelets. The normalised enrichment score (NES) represents the primary metric since it allows comparison between gene sets by taking the gene set size and correlations between the expression data and the gene sets into account [24]. Nom = Nominal, FDR = False Discovery Rate-adjusted.

observed increased aggregation after stimulation with arachidonic acid in our study may point to a decreased efficacy of acetylsalicylic acid in patients with sepsis.

Overall, the present data suggest that levels of reticulated platelets may become elevated due to various reasons—such as increased consumption or reactive thrombopoiesis—but their transcriptomic alterations will rather depend on the underlying disease condition. Further studies examining sorted reticulated platelets or platelet ribosome profiling in disease conditions different from profound, systemic inflammation may shed further light on disease-specific translation in platelets. In the long term, selective inhibition of platelet-specific translation could represent an attractive therapeutic target, provided that such translation has relevant disease-modifying impact.

## Limitations

One major limitation of the present study is its limited sample size. Thus, our results should be seen as hypothesis-generating and specifically verified in larger cohorts. Yet, our study reports to date the largest number of individual RNA-sequencing datasets from platelets in patients with sepsis. Also, according to sample size estimation, the current number of samples may detect transcriptomic changes to the described extent (log2 fold change of >1.5) at a power of 0.9. Still, as it is not unusual with clinical samples as compared to samples from standardised experimental conditions, we observed considerable heterogeneity between the samples that could not be explained by available patient characteristics. As mentioned above, we did not perform ribosomal foot printing to specifically assess platelet translation. Furthermore, we did not perform the VASP assay or cytometric determination of leukocyte-platelet aggregates which may have identified alterations of platelet activation in response to non-aggregating stimulation conditions.

## Conclusions

In patients with sepsis, increased levels of immature platelets and decreased levels of putative circular RNA transcripts in platelets point to a less degraded platelet transcriptome. Comprehensive analysis of genes regulated in platelets from patients with sepsis suggests increased protein translation that is characteristic for profound, systemic inflammation.

## Supporting information

**S1 Fig. Percentage of PCR duplication in individual samples.** PCR duplication as identified and removed by Je-MarkDuplicates in individual samples. P value from Mann-Whitney U test.
(TIF)

**S2 Fig. Number of expressed transcripts per individual patient.** For each individual patient, the number of transcripts with a normalized count of $\geq 10$ is displayed according to their gene biotype.
(TIF)

**S3 Fig. Functional annotation of differentially expressed genes.** Functional annotation of differentially expressed genes is analysed by ClueGO, a Cytoscape plugin, for upregulated genes (A) and downregulated genes (B) with similar parameters using Gene Ontology (GO), Reactome and KEGG terms. Terms with highest significance within one network are spelled in the respective colour.
(TIF)

**S1 Table. Results from Gene Set Enrichment Analysis for pre-defined gene sets related to platelet activation and platelet inhibition.** Results from Gene Set Enrichment Analysis for 29 gene sets retrieved from Molecular Signature Database related to platelet activation or platelet inhibition, according to [23], for possible enrichment in platelets from patients with sepsis. (XLSX)

**S2 Table. Results from DeSeq2 including normalised counts for individual samples.** Sheet 'all transcripts' depicts DeSeq2 results and normalised counts for all annotated transcripts. Sheet 'differentially exp transcripts' depicts DeSeq2 results and normalised counts for transcripts with adjusted p value < 0.05 and FC > 1.5. (XLSX)

## Acknowledgments

The skilful technical assistance of Cornelia Ebel and Lucas Gegalski is greatly acknowledged.

## Author Contributions

**Conceptualization:** Thomas G. Nührenberg, Franz-Josef Neumann, Christian Stratz, Marco Cederqvist, Willibald Hochholzer.

**Data curation:** Jasmin Stöckle, Christian Stratz, Marco Cederqvist.

**Formal analysis:** Thomas G. Nührenberg, Jasmin Stöckle, Federico Marini, Björn A. Grüning, Marco Cederqvist.

**Supervision:** Lutz Hein.

**Visualization:** Thomas G. Nührenberg, Mark Zurek.

**Writing – original draft:** Thomas G. Nührenberg, Marco Cederqvist.

**Writing – review & editing:** Thomas G. Nührenberg, Jasmin Stöckle, Federico Marini, Mark Zurek, Björn A. Grüning, Vladimir Benes, Lutz Hein, Franz-Josef Neumann, Christian Stratz, Marco Cederqvist, Willibald Hochholzer.

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
