## [Decision Letter · Decision Letter 0]

1 Jul 2021

Impact of high platelet turnover on the platelet transcriptome: Results from platelet RNA-sequencing in patients with sepsis 

Dear Dr. Nührenberg,

We're writing to let you know that we've checked your manuscript for our basic formatting requirements and we're now sending it to a staff editor for checks related to our editorial policies. Once this review is complete we will assign your submission to an Academic Editor for peer review. This is part of our standard process for submissions and no action is required from you at this time.

For your reference, our submission guidelines can be found on this page of our website: http://journals.plos.org/plosone/s/submission-guidelines.

Thank you for submitting your work to PLOS ONE and supporting our mission of Open Science.

Kind regards,

Irene Nathalie Fernandez Tolentino

PLOS ONE

---

## [Decision Letter · Decision Letter 1]

5 Nov 2021

Impact of high platelet turnover on the platelet transcriptome: Results from platelet RNA-sequencing in patients with sepsis

PONE-D-21-17284R1

Dear Dr. Nührenberg,

We’re pleased to inform you that your manuscript has been judged scientifically suitable for publication and will be formally accepted for publication once it meets all outstanding technical requirements.

Kind regards,

Katherine James, Ph.D.

Academic Editor

PLOS ONE

Additional Editor Comments (optional):

Reviewers' comments:

Reviewer's Responses to Questions

**Comments to the Author**

1. If the authors have adequately addressed your comments raised in a previous round of review and you feel that this manuscript is now acceptable for publication, you may indicate that here to bypass the “Comments to the Author” section, enter your conflict of interest statement in the “Confidential to Editor” section, and submit your "Accept" recommendation.

Reviewer #1: All comments have been addressed

Reviewer #2: All comments have been addressed

Reviewer #3: All comments have been addressed

2. Is the manuscript technically sound, and do the data support the conclusions?

Reviewer #1: Yes

Reviewer #2: Yes

Reviewer #3: (No Response)

3. Has the statistical analysis been performed appropriately and rigorously? 

Reviewer #1: Yes

Reviewer #2: Yes

Reviewer #3: (No Response)

4. Have the authors made all data underlying the findings in their manuscript fully available?

Reviewer #1: Yes

Reviewer #2: Yes

Reviewer #3: (No Response)

5. Is the manuscript presented in an intelligible fashion and written in standard English?

Reviewer #1: Yes

Reviewer #2: Yes

Reviewer #3: (No Response)

6. Review Comments to the Author

Reviewer #1: (No Response)

Reviewer #2: Well done revisions -- this is a very nice work and now very broadly visible in PLOS, congratulations

Reviewer #3: (No Response)

7. PLOS authors have the option to publish the peer review history of their article (what does this mean?). If published, this will include your full peer review and any attached files.

Reviewer #1: No

Reviewer #2: No

Reviewer #3: No

---

## [Editor Report · Acceptance letter]

10 Nov 2021

PONE-D-21-17284R1 

Impact of high platelet turnover on the platelet transcriptome: Results from platelet RNA-sequencing in patients with sepsis 

Dear Dr. Nührenberg:

I'm pleased to inform you that your manuscript has been deemed suitable for publication in PLOS ONE. Congratulations! Your manuscript is now with our production department. 

Kind regards, 

on behalf of

Dr. Katherine James 

Academic Editor

PLOS ONE